# Tool Use in Horses

**DOI:** 10.3390/ani12151876

**Published:** 2022-07-22

**Authors:** Konstanze Krueger, Laureen Trager, Kate Farmer, Richard Byrne

**Affiliations:** 1Department Zoology/Evolutionary Biology, University of Regensburg, Universitätsstraße 31, 93053 Regensburg, Germany; 2Department Equine Economics, Faculty Agriculture, Economics and Management, Nuertingen-Geislingen University, Neckarsteige 6-10, 72622 Nürtingen, Germany; laureen.esch@gmx.de; 3Department of Animal Welfare, Ethology, Animal Hygiene and Animal Husbandry, Veterinarian Medicine, Ludwig Maximilian University Munich, Veterinärstr. 13/R, 80539 München, Germany; 4Centre for Social Learning & Cognitive Evolution, School of Psychology, University of St Andrews, St Andrews KY16 9JP, Scotland, UK; katefarmer74@gmail.com (K.F.); rwb@st-andrews.ac.uk (R.B.)

**Keywords:** crowdsourcing, horse, innovation, mule, management, tool use

## Abstract

**Simple Summary:**

Tool use has not yet been confirmed in horses, mules or donkeys. As this subject is difficult to research with conventional methods, we used crowdsourcing to gather data. We asked equid owners and carers to report and video examples of “unusual” behaviour via a dedicated website, and we searched YouTube and Facebook for videos of equids showing tools. From 635 reports, including 1014 actions, we found 13 unambiguous cases of tool use. Tool use was associated with restricted management conditions in 12 of the 13 cases, and 8 of the 13 cases involved other equids or humans. The most frequent tool use, with seven examples, was for foraging, for example, equids using sticks to scrape hay into reach. There were four cases of tool use for social purposes, such as horses using brushes to groom others, just one case of tool use for escape, in which a horse threw a halter when it wished to be turned out, and one case of tool use for comfort, in which a horse scratched his abdomen with a stick. Equids therefore can develop tool use, especially when management conditions are restricted, but it is rare.

**Abstract:**

Tool use has not yet been confirmed in horses, mules or donkeys. As this subject is difficult to research with conventional methods, we used a crowdsourcing approach to gather data. We contacted equid owners and carers and asked them to report and video examples of “unusual” behaviour via a dedicated website. We also searched YouTube and Facebook for videos of equids showing tool use. From 635 reports, including 1014 behaviours, we found 20 cases of tool use, 13 of which were unambiguous in that it was clear that the behaviour was not trained, caused by reduced welfare, incidental or accidental. We then assessed (a) the effect of management conditions on tool use and (b) whether the animals used tools alone, or socially, involving other equids or humans. We found that management restrictions were associated with corresponding tool use in 12 of the 13 cases (*p* = 0.01), e.g., equids using sticks to scrape hay within reach when feed was restricted. Furthermore, 8 of the 13 cases involved other equids or humans, such as horses using brushes to groom others. The most frequent tool use was for foraging, with seven examples, tool use for social purposes was seen in four cases, and there was just one case of tool use for escape. There was just one case of tool use for comfort, and in this instance, there were no management restrictions. Equids therefore can develop tool use, especially when management conditions are restricted, but it is a rare occurrence.

## 1. Introduction

Tool use has been described in a wide range of animal species [1], meeting the definition “… the external employment of an unattached environmental object to alter more efficiently the form, position, or condition of another object, another organism, or the user itself when the user holds or carries the tool during or just prior to use and is responsible for the proper and effective orientation of the tool”. In horses, mules and donkeys, tool use has not yet been demonstrated, even though it may be considered likely [2] as equids are mostly kept under human management and tool use has been reported more frequently in captive animals [3,4].

That horses may have the ability to use tools is suggested by evidence of their innovative abilities. Equids developed innovative solutions for handling complicated feeders [5] and for dealing with environmental restrictions in foraging, movement and social contact [6,7,8,9,10], for example, by opening locked doors and gates and by harvesting apples by kicking the trees [2,11]. Others developed “less immediately functional” innovations for play and comfort, such as playing with sticks and piling up soft bedding for resting and sleeping [2]. Some of the innovations were solitary and served the innovator itself, but others were social and were directed towards conspecifics and humans [2]. The observation of innovative social behaviour is in line with prior demonstrations of horses using symbols for heterospecific communication, that is to say, communication with another species, when requesting humans to put on or take off a rug [12], or for prompting people to open gates to allow access to feed [13,14].

The lack of reports of tool use in equids may simply reflect the phenomenon’s rarity [3,4]. Evidence for the existence of such rare behaviour may be obtained from crowdsourcing [15]. For example, previous crowdsourcing studies have analysed the range of flexibility of animal problem-solving abilities [16], cognitive capacities in goats [17], door opening techniques, and innovative behaviour in horses [2,11]. Several methods have been used. Some amassed reports retrospectively, submitted by specialist researchers or enthusiasts, including studies on birds [18,19], primates [20,21,22], elephants [15], dogs [23], horses [11,24] and general wildlife [25]. Efforts need to be made to exclude reports that do not meet the definitions [19,26].

Others have searched journals for keywords such as “unusual” or “novel” [18,19,22,27]. A third approach is to ask trained personnel and researchers for contemporary reports [15]. A fourth has been to search internet platforms such as YouTube or Facebook for video material of animal behaviour, as applied in a study on human responses to tail chasing in dogs [28], play in dogs and horses [26], and door opening and innovations in horses [2,11]. If videos with unclear or manipulated content are excluded, and layperson documentations are available that clearly demonstrate that films have not undergone any postproduction editing, YouTube and Facebook videos can provide very useful raw footage [26,29].

Data collection of this kind runs the risk of compiling false or unrepresentative reports, thereby generating a biased data set [19,30,31]. In addition, responses may be biased by the over-representation of reports from highly motivated respondents [32], reports about socially desirable items (such as a “clever animal”) or even the respondents’ moods [30]. However, the approach has advantages that may offset its deficiencies. It potentially provides a large data set of rare observations [3,4], which could not possibly be collected by a single research team engaged in experimentation [4,15,18,25,26,29,33]. A large sample size increases the credibility of reports [15] especially when data such as pictures or videos are available [29,33,34].

In our study, we contacted equid owners and caretakers directly and via the internet and asked them to observe and document unusual behaviour by video and questionnaire at the website (https://innovative-behaviour.org accessed on 10 July 2022). In addition, we searched the internet platforms YouTube and Facebook for videos of equids showing tool use [2,11]. Among 635 reports, which collectively described or depicted 1014 behaviours, we found 13 cases that were in line with the definition of tool use [1].

In addition to discovering whether equids used tools [1,35], we also hoped to clarify the conditions which promote the development of tool use. We asked (a) whether certain management situations favour the development of tool use in equids [2] and whether (b) individual equids use tools solitarily, socially in an interspecific context with other equids, or in a heterospecific context with humans [12,13,14].

## 2. Materials and Methods

### 2.1. Study Location, Website, and Videos

Horse, mule, and donkey owners and caretakers were asked to report on “unusual”, novel behaviour through a website (https://innovative-behaviour.org accessed on 10 July 2022), via horse journals, Facebook, various private websites, and at conferences and public talks in Germany, Austria, France, Hungary, Switzerland, the UK, and the U.S.A., between July 2012 and December 2021. Reports were submitted in English, German, or French. We used a quantitative–qualitative mixed questionnaire approach as described in detail by [2], Krueger et al. (2021 a). In addition, we collected unusual behaviours, and environmental and individual specific aspects, shown on video material from the internet platforms YouTube and Facebook with the keywords “clever”, “smart”, “unusual”, “play”, “open door”, “open gate”, “escape”, “run away”, “horse”, “pony”, “donkey”, and “mule”. The complete resulting data set is available in previous publications on door and gate opening behaviour [11] and on innovative behaviour in equids [2].

### 2.2. Cases and Case Selection

Tool use (e.g., Figure 1) was reported for 20 equids (Appendix A), mostly as a side observation of 635 reports, which collectively described or depicted 1014 innovative behaviours [11].

Contentious cases were excluded from the total data set as described in [2]. The 20 “tool use” cases were rated by three independent observers, one professor, one PhD student and one bachelor’s student in equine science, on whether they were unambiguous. The observers agreed in 100% of the cases (inter-observer agreement: Cohen’s Kappa κ = 1) and found 13 cases to be unambiguous (Figure 1; Table 1 and Appendix A) and seven ambiguous (Appendix A). The seven ambiguous cases were excluded from further analysis because they may have reported actions that were not innovative tool use. They may have been:
(a)About trained behaviour (*n* = 1): one video clearly showed that the behaviour was aided and reinforced verbally and with feed (Appendix A).(b)Possibly the result of reduced welfare (*n* = 1), but not a solution for the underlying deficiency [9,10]: for example, one horse showing repetitive, stereotypic behaviour when scraping the ground with a stick (Appendix A).(c)Did not fit the definition [1] used in the study (*n* = 5), because the manipulated objects were not detached from the environment (Appendix A).

The remaining 13 cases (Table 1; Figure 1; Appendix A) were chosen for investigation. Nine owners reported that they did not encourage the equids to show the behaviour, two gave food as part of regular feeding routines after the behaviour was shown and in two cases the reaction of the persons was unknown.

### 2.3. Animals

The animals were 13 domestic equids, comprising 1 mule and 12 horses. They comprised one female, nine castrated males, one uncastrated male, and two equids for which the sex could not be discerned. The mean age at which equids were reported to have started showing the behaviour was 6.5 years (median, min. = 0.5, max. = 14). The horses were of various breeds and these were summarised according to the breed types deployed in genetic studies [36,37]: thoroughbred horses (*n* = 1), Arabian horses (*n* = 3), and warmblood horses (*n* = 6). In two cases the breed type was not reported or was not obviously visible in the videos. Animal characteristics which were not reported or clearly visible in videos were not considered for the analysis (Appendix A).

From the case descriptions and the videos, we identified the housing and management conditions: nine equids were kept in group stabling and three in single box stabling (one unknown), nine had unrestricted access to pasture and two restricted access to pasture (two unknown), nine had unrestricted contact to other equids and two restricted social contact (two unknown), five received unrestricted roughage and six restricted roughage (two unknown).

### 2.4. Behavioural Categories

Tool use cases were assigned to four behavioural categories developed from the immediate, observable context in which the behaviour was shown. We classified the categories as escape (*n* = 1), foraging (Figure 1; *n* = 7), comfort (*n* = 1) and social (*n* = 4) [2] (Table 1).

Seven equids showed their respective tool use more than 20 times, four showed it 2–10 times, and in two cases the frequency of the behaviour was unknown. Apart from tool use, some equids showed a median of 1 (min. = 0, max. = 4) further behaviours that were classified as “innovative“ [2] (Appendix A).

From the case reports or the videos, we concluded whether the tool use was solitary (was only for use of the tool using animal itself and did not include any other individual; Figure 1) or was interspecific (included another equid) or heterospecific (included a person; Table 1).

### 2.5. Data Analysis

R Studio (version 0.99.484, Boston, MA, USA) [38] of the R-Project statistical environment (R Development Core Team, 2022) and the Rcommander (package Rcmdr) were used for statistical analysis and the depiction of the data. Most of the data were not normally distributed (Shapiro–Wilk test). Likelihood equations were calculated with chi-squared tests for ordinal data and with binomial tests for binomial data (Appendix A). A Spearman rank correlation test was applied to compare the variance of management data and tool use behaviour categories (Appendix A). The significance level was set at 0.05 and all tests were two-sided.

## 3. Results

Among the equids reported displaying “unusual behaviour” only 13 out of 1014 showed unambiguous tool use. Equids showing tool use were mostly castrated males (chi-squared test: *n* = 11, χ = 11.636, df = 2, *p* = 0.003), rather than females (binomial test: *n* = 10, *p* = 0.02) or uncastrated males (binomial test: *n* = 10, *p* = 0.02; Appendix A).

Restrictions in the equids’ management conditions correlated with the behavioural category under which we recorded the tool use: horses with restrictions in free movement showed tool use for escape, equids with restrictions in access to feed primarily displayed tool use for foraging (Figure 1), and horses with restrictions in social contact displayed social tool use (Spearman correlation test: *n* = 13, rs = 0.708, *p* = 0.01; Appendix A). The only case which we categorized as tool use for comfort was displayed by a horse that lived under conditions which covered all basic needs.

Tool use reported for enhancing comfort (*n* = 1) was solitary. Tool use for foraging (*n* = 7) was solitary in most cases (Figure 1; *n* = 4), but in one case included a mule, and in two cases humans. Tool use which served social purposes (*n* = 4) was interspecific (*n* = 3), i.e., included other horses, or heterospecific (*n* = 1), i.e., included humans. The only escape tool use reported (*n* = 1) was heterospecific, the tool used to enhance free movement included a person (Figure 2).

## 4. Discussion

Crowdsourced reports and videos provided 13 descriptions of tool use [1,35], in 12 horses and 1 mule. To our knowledge, this is the first time that tool use in equids has been scientifically described and supports the suggestion that rare instances of tool use in other “non-tool using species” may be found in the future [3,4].

Equids may develop tool use for various reasons, the most common being to improve their situation when management conditions are restricted. Some displayed solitary tool use and may have done so to enhance the fulfilment of their own needs, such as free movement, comfort and their feeding conditions [10]. Others may have used tools to enrich their social situation [10]. Some tool use for feeding, and all tool use in social situations, included other equids (was interspecific) or humans (was heterospecific). Tool use that included other individuals may have served to communicate insufficiencies in basic needs [10], comparable to findings in equids that displayed referential heterospecific communication [12,13,14], when asking humans to take off or put on a rug, or when asking for access to feed.

Communicating needs by using tools requires a more complex mental level, compared to direct actions such as pointing with the head at desired feed [13,14]. When a horse indicates a wish by directing the attention of a person towards a desired feed item by body movements, it demonstrates an understanding of the direction of the person’s attention [39] and in the person understanding the pointing behaviour [14]. In addition, when a horse appears to communicate a need to be led out of the stable by throwing a halter, which is used for leading, or a need for feed by throwing an empty feed bucket, in front of an approaching person, the horse displays an understanding of communal belief [40], i.e., the communal understanding that a halter needs to be put on to leave the stable and a bucket needs to be filled for feeding.

However, tool use in equids may not primarily be prompted by resource shortage, as was concluded for innovative behaviour in a feeding and escape context [2,24]. Favourable living conditions, in which all the needs of the equids are covered [10], may also encourage the development of tool use, as shown for most innovations in comfort and play situations [2]. Play may also prompt the development of tool use [41]. This is supported by the finding that more males than females used tools in the present study and male horses have been reported to play more than females [42,43]. The small sample size of tool use to improve comfort in the present study does not allow for robust conclusions on the reason for tool use development in equids and so remains mostly descriptive.

Interestingly, in contrast to the single case of tool use for escape in the present study, escape, rather than foraging, social purposes or comfort was reported to be the main context for equids to develop generally innovative solutions [2]. Furthermore, equids may display oral tool use in most cases because it is difficult to perform delicate manipulations with hoofs, and this may result in tool use for foraging more often than for other purposes, as in the present study.

### Potential Biases in the Data

It is important to acknowledge potential biases in our data. Collecting data with crowdsourcing methods may introduce biases into the data set. We took care to exclude unreliable or biased reports [19,26], one possibly on reinforced (i.e., trained) behaviour and one on reduced welfare [9,10,25], and supported the reports with pictures and videos [2,22,28,29,33]. This allowed data to be amassed with some replications of very rare observations [30] and allowed a cautious approach to the analysis of the circumstances of tool use [15,18,25,26,29,33].

In all behaviour categories, present and previous owners may have unintentionally reinforced the behaviour by rewarding the animal with enhanced affection [31]. Therefore, the present study applied two direct and two catch questions [24] to filter reports of unintentionally trained behaviour by present owners. The catch questions asked whether the equids received feed in the immediate context of showing the behaviour or whether the present owners were pleased when they observed the behaviour and may, therefore, have provided unintentional reinforcement in the form of increased attention or positive reactions, of which the horse previously learned to be followed by rewards. However, unintentional reinforcement of present owners or training by previous owners might not have been obvious in all cases.

In addition, it is important to acknowledge the limitations of crowdsourcing methods, as they may affect the interpretations. Where we find that a factor covaries with tool use frequency, that could be a genuine finding or an artefact generated by bias in the respondent’s behaviour or reporting. For instance, the lack of observations of tool use in the context of escape may be a genuine finding. It may also reflect a biased tendency of respondents to keep equids in pleasant management conditions where animals do not escape. Furthermore, bias may arise because respondents were pleased to detect and report tool use for foraging, comfort and social communication more enthusiastically [31], relative to equid owners keeping their horses in conditions that prompted their escape. In addition, equids may apply tool use for escape mostly when they are unobserved. Owners intervene to prevent escape if they see it happening, because it may result in serious accidents [44,45], and horses may be punished for trying to escape. Therefore, learning to use a tool to help them escape may be something horses do when no one is around. Access to data on management conditions amongst all equid owners would have been ideal. This would allow statistical analyses to quantify the level of bias. While that was not possible for the current study, it may remain a possibility for future studies.

It is unknown whether such biases exist, and, if so, their magnitude. Therefore, we take our findings at face value and provide interpretations that would be appropriate for an unbiased data set. However, we stress that until such a time as the level of bias can be quantified, our findings should be regarded as provisional and suggestive rather than definitive.

## 5. Conclusions

We conclude that horses have the potential to develop behaviour involving tool use, particularly to improve their situation when management conditions are restricted, for example for foraging and improving social contact; however, this remains a rare phenomenon.

## Figures and Tables

**Figure 1 animals-12-01876-f001:**
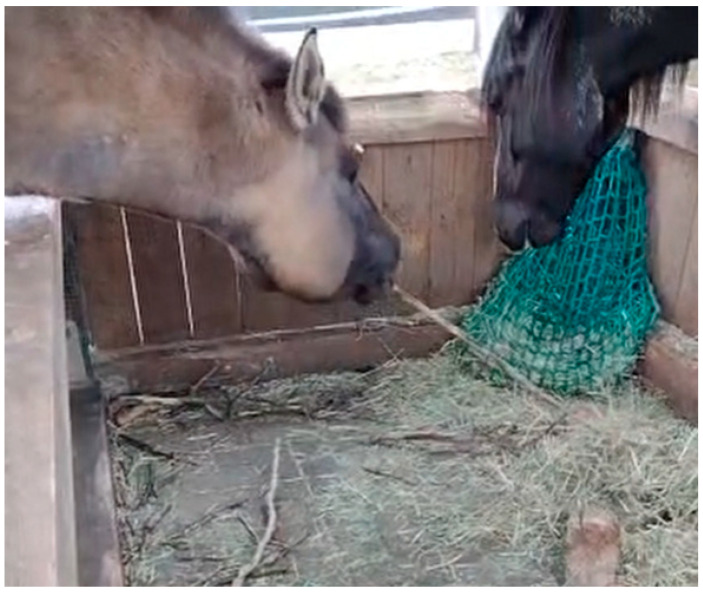
Horse using a stick to scrape hay into reach (Appendix A).

**Figure 2 animals-12-01876-f002:**
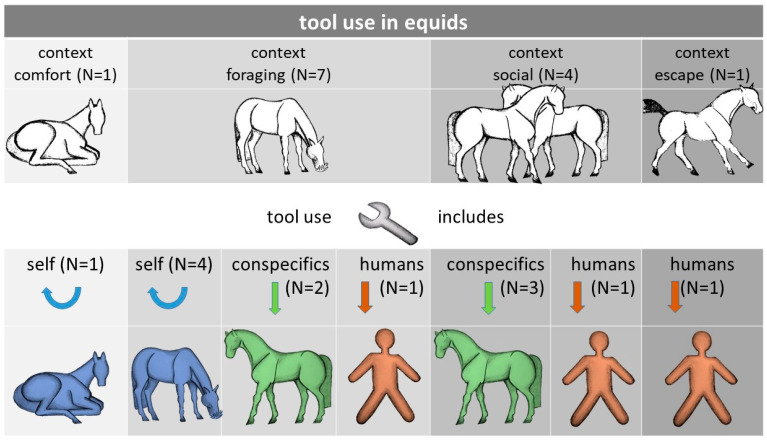
Tool use context and direction in horses. Blue colouration indicates that tool use was solitary (*n* = 5), green colouration that tool use included a conspecific (*n* = 5), and brown colouration that it included humans (*n* = 3).

**Table 1 animals-12-01876-t001:** Tool use in equids categorized according to its direction: interspecific = directed towards other equids (brown), heterospecific = directed towards humans (green), solitary = for the tool using animal itself (blue).

Type	Description	Sex	Restricted Access to	Behaviour Category
Interspecific (involving equids) Heterospecific (involving people) Solitary		Mare (M) Gelding (G) Stallion (S)	Pasture (P) Equids (E) Roughage (R)	Social Forage Escape Comfort
Interspecific	Horse A uses large stick to chase horse B and make B move faster.	G	Unrestricted	Social
Interspecific Appendix A	Horse A takes bucket in its mouth and swings it at horse B, who is eating from a full bucket. A continues until B retreats and A can eat from the full bucket.	Unknown	Unknown	Forage
Interspecific	Horse picks up brushes and tries to groom other horses.	G	R	Social
Interspecific Appendix A	Mare and foal in a box, foal grooms mother with comb.	Unknown	Unknown	Social
Heterospecific	Horse picks up a stick and swings it at person.	G	R	Social
Heterospecific	Horse bangs bucket against wall to demand feed, When on a low fenced paddock, he throws the bucket over the fence towards the feed room.	S	Unrestricted	Forage
Heterospecific	At feed time, horse throws the empty feed bucket in front of the owner. When she fills the bucket, the horse drags it into his stable.	G	E	Forage
Heterospecific Appendix A	Horse grabs the halter next to the box and throws it at the owner’s feet. Behaviour developed when was on box rest and so had restricted movement, forage and social contact.	G	P + E	Escape
Solitary Appendix A	Horse picks up stick and scratches abdomen/belly with it.	G	unrestricted	Comfort
Solitary	Horse uses a stick to rake hay out from the hay rack when hay is almost empty. Other horses observe and eat from the raked hay.	G	R	Forage
Solitary Appendix A	Horse takes a stick in its mouth and uses it to rake hay from under the hay rack. Horse brings stick in from outside the stable for this.	G	R	Forage
Solitary Appendix A	Mule takes a stick in his mouth and rakes hay from underneath the hay rack, after observing a horse doing the same and eating the raked hay. Mule started raking hay when the horse was removed from the group.	G (Mule)	R	Forage
Solitary Appendix A	The horse takes a stick in her mouth and rakes hay within reach. She eats the hay, then searches for the stick again, and repeats behaviour.	M	P + R	Forage

## Data Availability

The data presented in this study are available in the Appendix A.

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
