# Peer review of "Tool Use in Horses"

_animals, 2022, doi:10.3390/ani12151876_

Round 1

Reviewer 1 Report

 Interesting compilation of examples.Equine is an adjective,. Say horse or if mules and donkeys included say Equus.  The use of a lead rope to scoop food into the stall was most interesting.

1)The main question was " do horses use tools?"

2)the topic is relevant

3) its adds to the subject area, which has very little at this point

4) the methodology-using videos provided by owners-is novel

 There aren't too many improvements the authors could make except to acknowledge that the horse may have been taught the tool use rather than the horse spontaneous doing so. Even if the present ownmers are honest, a previous owner may have taught the horse.

5 the conclusions are consistent

 6 the references are appropreiate

Author Response

  • Dear referee,

Thank you so much for your encouraging review. We will work on your suggestions point by point below.

Comments and Suggestions for Authors

 Interesting compilation of examples. Equine is an adjective,. Say horse or if mules and donkeys included say Equus. 

  • The point is well taken. "Equine" is both an adjective and a noun. Collins defines “equine” as an adjective and uncommon adjective-noun, Merriam-Webster lists is as equally noun and adjective. Equus is a genus, and refers to the genus, not the animals, so can't be used in the context suggested. We suggest using "equid", and have changed the terminology throughout the manuscript.

The use of a lead rope to scoop food into the stall was most interesting.

  • We agree, and would have loved to include this example as unambiguous. However, in the video a person provides significant verbal reinforcement.

1)The main question was " do horses use tools?"

2)the topic is relevant

3) its adds to the subject area, which has very little at this point

4) the methodology-using videos provided by owners-is novel

 There aren't too many improvements the authors could make except to acknowledge that the horse may have been taught the tool use rather than the horse spontaneous doing so. Even if the present owners are honest, a previous owner may have taught the horse.

  • Good point, we included this notion into the discussion.

5 the conclusions are consistent

 6 the references are appropreiate

Reviewer 2 Report

This is a nicely presented report of cleverly sourced and evaluated material as evidence of tool use in horses. The introduction and discussion are scientifically sound with appropriately discussed limitations. 

This work provides organized evidence for what has been discussed informally among equine behavior scientists for decades and what I have seen a few times myself over a very long career. One quite memorable anecdotal I often discuss with students was a situation of two young foals living under natural social and environmental conditions.  They were playing when one picked up a stick, then "used it" to poke the playmate, multiple times during a play chase, then seemingly intentionally dropping the stick and the playmate then picking up the stick and poking the other.  They then had "some tug of war" type interaction with both foals appearing to try to get sole control of the stick.  I suppose there could be alternate explanations for this sequence and other situations I have witnessed, but so pleased to see you publishing more widespread examples with video evidence that can be reviewed and discussed regarding the nuances of tool use.

Minor questions/suggestions:

Line 45- I am surprised that you the 1980 definition of tool use that requires the "tool" to be unattached.  As long as it can manipulated to complete the task. An example, again from semi-feral animals, I have seen ponies when loafing under low branches,  grab a branch in mouth and pull and hold it down against their back and then rub back and forth, so like a back scratcher. I have also seen them on hot summer days when loafing in the shade of low trees and bushes, grab a low branch in their mouth and rhythmically pull up and down or back and forth seemingly to "fan."   (the effect/reinforcement may be chasing insects or cooling). Or we regularly see animals stand over a tall stiff weed or downed tree branch rocking back and forth scratching their underbelly.

S4 Example -The halter tosser.  This seems to be the weakest example. You might be reading too much into the scenerio. In these thwarted goal situations, horses often fidget with/manipulate anything handy. For example it could be a bucket, a barn implement. But I think it is a good for discussion of that point.

Line 209 Are you sure you want to declare tool use has never before been described in horses in scientific literature.  You may wish to soften that, perhaps "to your knowledge." Or "we are unaware of previous reports."

Line 255 - 261 In horses in a thwarted goal situation, any reaction of the owner, whether intended to be affection or a "pleasure" response, unless extremely punitive, is likely reinforcing. Any reaction/attention is better than nothing in those situations.

Author Response

  • Dear Referee,

Thank you very much for the very thoughtful comments and  very positive encouragement. We will consider and answer your suggestions point by point below.

Comments and Suggestions for Authors

This is a nicely presented report of cleverly sourced and evaluated material as evidence of tool use in horses. The introduction and discussion are scientifically sound with appropriately discussed limitations.

This work provides organized evidence for what has been discussed informally among equine behavior scientists for decades and what I have seen a few times myself over a very long career. One quite memorable anecdotal I often discuss with students was a situation of two young foals living under natural social and environmental conditions.  They were playing when one picked up a stick, then "used it" to poke the playmate, multiple times during a play chase, then seemingly intentionally dropping the stick and the playmate then picking up the stick and poking the other.  They then had "some tug of war" type interaction with both foals appearing to try to get sole control of the stick.  I suppose there could be alternate explanations for this sequence and other situations I have witnessed, but so pleased to see you publishing more widespread examples with video evidence that can be reviewed and discussed regarding the nuances of tool use.

  • Thank you for sharing this very interesting anecdote. It reflects very well what we found in other reports.

Minor questions/suggestions:

Line 45- I am surprised that you the 1980 definition of tool use that requires the "tool" to be unattached.  As long as it can manipulated to complete the task. An example, again from semi-feral animals, I have seen ponies when loafing under low branches,  grab a branch in mouth and pull and hold it down against their back and then rub back and forth, so like a back scratcher. I have also seen them on hot summer days when loafing in the shade of low trees and bushes, grab a low branch in their mouth and rhythmically pull up and down or back and forth seemingly to "fan."   (the effect/reinforcement may be chasing insects or cooling). Or we regularly see animals stand over a tall stiff weed or downed tree branch rocking back and forth scratching their underbelly.

  • Before submitting the manuscript, we actually discussed the aspect of presenting tool use of unattached objects (also sometimes referred to as border line tool use), as we did receive reports that were close to your description above. However, since this is the first publication describing tool use in horses we decided to present tool use of unattached objects only, as these cases are clear and well accepted by the scientific community. Integrating border line tool use may raise a debate, which finally results in rejection of the whole concept. We would leave this debate to follow up studies, which may discuss whether it is appropriate to widen the definition of tool use in horses.

S4 Example -The halter tosser.  This seems to be the weakest example. You might be reading too much into the scenerio. In these thwarted goal situations, horses often fidget with/manipulate anything handy. For example it could be a bucket, a barn implement. But I think it is a good for discussion of that point.

  • You may be right. However, as two more examples were reported in which horses threw buckets to communicate their wish to be fed, we felt that throwing a halter may very well be in the range of communicating the wish to leave the box. As this behaviour was reported to occur in particular situations of need, and only when persons that may help to cover this need were approaching, we decided to categorize them as feeding and escape tool us for heterospecific communication.
  • We added the feed bucket examples to the explanation of tool use for referential communication.

Line 209 Are you sure you want to declare tool use has never before been described in horses in scientific literature.  You may wish to soften that, perhaps "to your knowledge." Or "we are unaware of previous reports."

  • Very good suggestion, thank you. We added “to our knowledge”

Line 255 - 261 In horses in a thwarted goal situation, any reaction of the owner, whether intended to be affection or a "pleasure" response, unless extremely punitive, is likely reinforcing. Any reaction/attention is better than nothing in those situations.

  • Yes, this is correct. We tried to clarify the description of unintentional rewards as follows:

“In all behaviour categories, present and previous owners may have unintentionally reinforced the behaviour by rewarding the animal with enhanced affection [32]. Therefore, the present study applied two direct and two catch questions [25] to filter reports of unintentionally trained behaviour by present owners. The catch questions asked whether the equids received feed in the immediate context of showing the behaviour or whether the present owners were pleased when they observed the behaviour and may, therefore, have provided unintentional reinforcement in the form of increased attention or positive reactions, of which the horse previously learned to be followed by rewards. However, unintentional reinforcement of present owners or training by previous owners might not have been obvious in all cases.”